# Association between Prehospital Visits and Poor Health Outcomes in Korean Acute Stroke Patients: A National Health Insurance Claims Data Study

**DOI:** 10.3390/ijerph20032488

**Published:** 2023-01-31

**Authors:** Jinyoung Shin, Hyeongsu Kim, Youngtaek Kim, Jusun Moon, Jeehye Lee, Sungwon Jung, Rahil Hwang, Mi Young Kim

**Affiliations:** 1Department of Family Medicine, School of Medicine, Konkuk University, Seoul 05029, Republic of Korea; 2Department of Preventive Medicine, School of Medicine, Konkuk University, Seoul 05029, Republic of Korea; 3Department of Preventive Medicine, Chungnam National University Hospital, Daejeon 35015, Republic of Korea; 4Department of Neurology, National Medical Center, Seoul 04564, Republic of Korea; 5National Emergency Medical Center, National Medical Center, Seoul 04564, Republic of Korea; 6Department of Nursing, Fareast University, Eumseong 27601, Republic of Korea; 7Department of Nursing, College of Nursing, Shinhan University, Uijeongbu 11644, Republic of Korea; 8Department of Nursing, College of Nursing, Hanyang University, Seoul 15588, Republic of Korea

**Keywords:** stroke, delayed diagnosis, length of stay, patient readmission, mortality, administrative claims, health care

## Abstract

This study aimed to determine whether prehospital visits to other medical institutions before admission are associated with prolonged hospital stay, readmission, or mortality rates in acute stroke patients. Using the claims data from the Korean Health Insurance Service, a cross-sectional study was conducted on 58,418 newly diagnosed stroke patients aged ≥ 20 years from 1 January 2019 to 31 December 2019. Extended hospital stay (≥7 days; median value) following initial admission, readmission within 180 days after discharge, and all-cause mortality within 30 days were measured as health outcomes using multiple logistic regression analysis after adjusting for age, sex, income, residential area, and medical history. Stroke patients with a prehospital visit (10,992 patients, 18.8%) had a higher risk of long hospitalization (odds ratio = 1.06; 95% confidence interval = 1.02–1.10), readmission (1.19; 1.14–1.25), and mortality (1.23; 1.13–1.33) compared with patients without a prehospital visit. Female patients and those under 65 years of age had increased unfavorable outcomes (*p* < 0.05). Prehospital visits were associated with unfavorable health outcomes.

## 1. Introduction

Rapid and accurate detection of stroke is crucial for the timely initiation of treatment because strokes account for 11% of total deaths and 30% of disabilities requiring lifelong assistance [1,2]. Prolonged delays are classified as “prehospital delays” from symptom onset to hospital admission and “in-hospital delays” between patient arrival and initiation of treatment [3,4]. Prehospital delay is the most significant type of prolonged delay [5]. Prehospital delay is affected by patient-related factors (e.g., ability to recognize symptoms, awareness, or prior unfavorable experiences) and system-related factors (triage by an emergency medical service (EMS) or issues related to EMS accessibility including geographic (e.g., distance from a stroke care unit), temporal (e.g., traffic in “rush hour”), or socioeconomic factors (e.g., level of insurance coverage)) [6,7,8,9]. Existing guidelines recommend that an EMS be contacted immediately if a patient has stroke-related symptoms, ensuring the fastest transfer to the nearest stroke center [10,11]. However, half of stroke patients do not visit a proper hospital directly without using an EMS [12]. 

As measured by symptom onset-to-door time, early hospital arrival is known to affect health outcomes [5,10]. However, in a study of 539 Korean stroke patients who arrived at a hospital within 4.5 h of symptom onset, no difference in the hospital stay or mortality was observed compared with those who arrived after more than 4.5 h (average, ~40 h) [13]. Although early hospital arrival within six hours after stroke onset was associated with neurological improvement and positive functional outcomes in 6780 Japanese ischemic stroke patients, uncertainties in measuring the onset-to-door time have been raised. It may have been overestimated because the exact time of stroke onset was unknown and was instead defined as the last time the patient was known to be neurologically normal [14]. It is therefore necessary to identify a proper indicator that reflects prehospital delay, including related clinical outcomes. A prehospital visit, defined as visiting other medical institutions, not treatment sites, before diagnosis may be an objective and measurable indicator of prehospital delay, because Korean patients can choose their healthcare providers and institutions at every turn. 

To confirm the usefulness of prehospital visits as a measure of prehospital delay, this study was conducted to compare the health outcomes, such as extended hospital stays, readmission, and mortality, of newly diagnosed stroke patients according to the prehospital visits. 

## 2. Materials and Methods

### 2.1. Data Sources

A cross-sectional study was performed to evaluate health outcomes, such as length of hospital stay, readmission, and mortality, according to prehospital visit patterns in Korean stroke patients using the National Health Insurance Service (NHIS) claims data. The National Health Information Database is a public healthcare database maintained by the NHIS for health screening, sociodemographic variables, and mortality for the entire South Korean population (97.2% of NHIS insurers and 2.8% of medical aid) [15]. Data for more than 50 million individuals have been collected since 2002. The eligibility database includes information on income-based insurance contributions, demographic variables, and date of death. The National Health Screening Database provides information on health behaviors and clinical variables. The healthcare utilization database records the medical use of and prescriptions for both in- and outpatients with deidentified join keys. The healthcare provider database includes information on the type of institution, human resources, and equipment [15]. Claims data include diagnosis, procedures, and prescriptions identified by the International Classification of Diseases, Tenth Revision (ICD-10) codes, and Korean Drug and Anatomical Therapeutic Chemical Codes. 

### 2.2. Study Design

In the NHIS claims data, stroke inpatients aged ≥ 20 years who were admitted to the emergency department (ED) of secondary or tertiary hospitals and confirmed by brain computer tomography or magnetic resonance imaging from 1 January 2019 to 31 December 2019 (*n* = 101,033) were enrolled. After excluding participants who had a previous diagnosis of any stroke (*n* = 42,615), 58,418 patients remained in the study (Figure 1). 

A diagnosis of stroke involved at least one of the following ICD-10 codes in the NHIS data: I60 (subarachnoid hemorrhage), I61 (intracerebral hemorrhage), I62 (other nontraumatic intracranial hemorrhage), or I63 (cerebral infarction) without a previous diagnosis in the three years prior. Hemorrhagic stroke was classified as I60, I61, and I62; ischemic stroke was classified as I63. Patients with ICD-10 codes for hemorrhagic and ischemic strokes (*n* = 942) in 2019 were categorized according to the first code.

### 2.3. Study Variables

A prehospital visit was defined as a visit to a medical institution to treat conditions of I60 to I63 and I64 (stroke not specified as hemorrhage or infarction), I65 (occlusion and stenosis of a vertebral artery), I66 (occlusion and stenosis of a middle cerebral artery), I67 (other cerebrovascular diseases), or I68 (cerebrovascular disorder in diseases classified elsewhere) prior to admission to the ED of a secondary or tertiary hospital. 

Age was calculated based on participants’ birth year. Income status involving six groups (medical aid and the lowest quintile to the highest quintile) was obtained based on each patient’s insurance coverage. The residential area was classified into “high urbanization” areas, including administrative districts with a tertiary hospital or a stroke care unit, and “low urbanization” areas. Comorbidities, such as hypertension (I10), ischemic heart disease (I20, I24, and I25), congestive heart failure (I11, I13, and I50), diabetes mellitus (E10), and arrhythmia (I49 and I802), were assessed using a physician diagnosis.

An extended or “long” stay in a hospital was defined as any stay of more than seven days (median value of study participants’ hospitalization period) during the admission for initial stroke treatment. Readmission was defined as admission within 180 days of discharge. We investigated mortality within 30 days of stroke diagnosis, and there were no restrictions on the cause of death. 

### 2.4. Statistical Analysis

Continuous demographic variables were expressed as mean values and standard deviations, and categorical variables were expressed as frequencies and percentages. Age was presented as the mean with standard deviations, and categorical variables were used in the subgroup analysis. Univariate analysis was conducted between the prehospital-visit and no-prehospital-visit groups using t- and chi-square tests.

We calculated the odds ratio (OR) and 95% confidence interval (CI) of unhealthy outcomes, such as long stays in the hospital, readmission, and mortality, in the prehospital-visit group compared to the no-prehospital-visit group using multiple logistic regression analyses after adjusting for age, sex, income, residence area, and previous medical history (hypertension, ischemic heart disease, congestive heart failure, diabetes mellitus, and arrhythmia). We calculated the OR and 95% CI of unhealthy outcomes at the subgroup level of sex and age (divided at 65 years). The time interval between the prehospital visit and the day of admission for treatment was obtained from the claims data. All analyses were performed using SAS version 9.4 (SAS Institute Inc., Cary, NC, USA), and statistical significance was set at *p* < 0.05.

## 3. Results

### 3.1. Comparison of Demographic and Clinical Characteristics

The characteristics of the 58,418 stroke patients, including those with ischemic and hemorrhagic strokes, are shown in Table 1. Among them, 10,992 (18.8%) had visited a medical institution with stroke-related ICD codes. The mean age of the participants was 65.9 ± 14.6 years. Prehospital visits varied according to income status (*p* < 0.001). The low-income group with medical aid had a lower rate of prehospital visits than the group with medical insurance. Patients living in highly urbanized areas were more likely to be in the no-prehospital-visit group (*p* < 0.001). Patients with hypertension were more likely to be in the prehospital-visit group; however, patients with ischemic heart disease, congestive heart failure, diabetes mellitus, and arrhythmia were more likely to be in the no-prehospital-visit group (*p* < 0.001). The characteristics of the patients with ischemic and hemorrhagic strokes are presented in Appendix A.

### 3.2. Health Outcomes According to Prehospital Visits

The number of stroke patients with extended hospital stay was 35,456 (60.7%) (Table 2). The rate of extended hospital stay was higher in the prehospital-visit group than in the no-prehospital-visit group (62.5% vs. 60.2%, *p* < 0.001). Of the participants, 30,571 (52.3%) were readmitted, and 4372 (7.48%) died, with the prehospital-visit group exhibiting higher readmission (55.3% vs. 51.7%, *p* < 0.001) and mortality rates (7.94% vs. 7.38%, *p* = 0.043). 

After adjusting for age, sex, income, residence, and medical history (hypertension, ischemic heart disease, congestive heart failure, diabetes mellitus, and arrhythmia), the ORs and 95% CIs for extended hospital stay (OR = 1.06; 95% CI = 1.02–1.10), readmission (OR = 1.19; 95% CI = 1.14–1.25), and mortality (OR = 1.23; 95% CI = 1.13–1.33) in the prehospital-visit group were significantly higher than those in the no-prehospital-visit group (*p* < 0.001).

Adjusted for age, sex, income, residence, and medical history (hypertension, ischemic heart disease, congestive heart failure, diabetes mellitus, and arrhythmia). An extended hospital stay was defined as >7 days (median values). Readmissions over 180 days were recorded. Mortality within 30 days was recorded.

### 3.3. The Effects of Patient Characteristics on Health Outcomes of Prehospital Visits

In participants younger than 65 years, a prehospital visit was associated with an increased risk of an extended stay in the hospital (OR = 1.32; 95% CI = 1.24–1.41), readmission, and mortality (OR = 1.17; 95% CI = 1.03–1.32) compared with participants aged 65 years and older (OR = 0.90; 95% CI = 0.84–0.95; OR = 1.14; 95% CI = 1.08–1.21; and OR = 1.08; 95% CI = 0.97–1.19, respectively; both P interactions < 0.01, Figure 2). Female patients were at an increased risk of an extended hospital stay (OR = 1.21; 95% CI = 1.14–1.30), readmission (OR = 1.28; 95% CI = 1.19–1.36), and mortality (OR = 1.17; 95% CI = 1.05–1.31) compared to male patients (OR = 0.99; 95% CI = 0.93–1.05, OR = 1.10; 95% CI = 1.04–1.17, and OR = 1.08; 95% CI = 0.97–1.21, respectively; both P interactions < 0.05). 

Adjusted for age, sex, income, residence, and medical history (hypertension, ischemic heart disease, congestive heart failure, diabetes mellitus, and arrhythmia).

The associations between prehospital visits and poor health outcomes according to stroke type and patient characteristics, including income and urbanization, are presented in Appendix A.

## 4. Discussion

This study demonstrated that prehospital visit status was associated with an increased risk of extended hospital stay, readmission within 180 days, and 30-day mortality in newly diagnosed stroke patients. This association existed in patients with all types of strokes even after considering income, residence area, and comorbidities that may affect medical use behavior.

Previous studies on prehospital delay focused on measuring onset-to-door time to identify the beneficial effects of early treatment, such as reperfusion therapy, anticoagulants, or antiplatelet agents in ischemic stroke patients [16,17,18,19]. Studies show that arrival time at the hospital is associated with clinical outcomes, regardless of treatment or stroke severity. An Italian cohort study of 1847 ischemic stroke patients without reperfusion therapy found that arrival within two hours of stroke onset was associated with a decreased risk of 30-day mortality [20]. Early hospital arrival within six hours after stroke onset was associated with favorable functional outcomes in 6780 Japanese patients with ischemic stroke who did not receive reperfusion treatment or had only minor symptoms [14]. In this study, we did not identify the factors that explained why the prehospital-visit group had a high mortality rate. Acute stroke patients with a prehospital visit may have allowed the late initiation of general supportive care or treatment for a complication, which may have contributed to increased unfavorable health outcomes [14]. Additionally, a long hospitalization period or readmission may result in poor clinical prognosis and lead to increased mortality.

Presumptive onset based on symptoms, an indicator commonly used by many clinicians, may not accurately recognize the initial symptoms [21]. Our study found that delayed arrivals with a prehospital visit were associated with unfavorable outcomes, including an extended hospitalization period, a high readmission rate, and a high 30-day mortality rate in acute stroke patients, regardless of stroke type and treatment. Therefore, prehospital visits may be a predictor of health outcomes in patients with acute stroke. 

Among the prehospital-visit group (*n* = 10,992) in this study, 10,025 (91.2%) visited one medical institution and 967 (8.8%) visited two or more medical institutions. The proportion of patients who had made a prehospital visit on the same day as the final diagnosis was 8651 (78.7%). The number of patients who visited the stroke center one day later was 1198 (10.9%), and that for a visit two or more days later was 1143 (2–6 days; 10.4%). Prehospital delays of two days in 10% of cases is a relatively high rate, even with the limitation of counting one day after midnight in the claims data. Although the reason or required time for the prehospital visits is still being determined from these data, it may be that the prehospital delay was due to the number of medical institutions visited.

Prehospital delay can be divided into three steps: (1) symptom onset motivates a decision to seek medical attention, (2) time of the decision to seek medical attention to first medical contact, and (3) first medical contact to hospital arrival [21]. Steps 1 and 2 can be defined as personal recognition of early stroke symptoms by the patient, family, or observers and can be improved by public health education [21]. Step 3 is based on pre- and posthospital presentation delays. Efforts to recognize stroke symptoms earlier through public campaigns or education might help reduce prehospital delays. Based on our research, we emphasize the need for education to go directly to the ED or stroke care units. Patients unable to identify their symptoms initially might contact their local or community doctors rather than go directly to the ED [22].

The prehospital visit rate was higher in males over 65 years and with comorbid diseases than in females under 65 years with no comorbid conditions (Table 1). In this study, male patients had a higher rate of prehospital visits; however, poor health outcomes were more evident in female patients. Previous research points to sex differences as vulnerabilities in female patients regarding poststroke variables, such as muscle strength, psychological status (depressive mood or fatigue), and social status (e.g., a higher rate of living alone in females) [23,24,25]. Patients under 65 years of age are likely to underestimate their clinical presentation [5]. Therefore, they may be unable to determine whether they should visit the stroke units quickly, although we did not survey their initial symptoms. Patients without comorbid diseases may have a higher rate of prehospital visits; patients with comorbidities may have been alerted or educated regarding their comorbid diseases. However, hypertension has such a high comorbid rate that it is thought that this effect did not appear. 

In Korea, a relatively low barrier to medical use may have influenced the rate of prehospital visits because of fewer financial obstacles. We found that high-income status was associated with an increased risk of extended hospital stay and readmission among ischemic stroke patients, as suggested in Appendix A. The cost of medical insurance is set according to income level, but the medical services provided are the same regardless of the income level. An individual’s income level can affect the hospitalization duration in Korea’s medical payment system. Therefore, the generalizability of the results must be validated in other cohorts, especially in other countries. 

Compared with high urbanization areas, acute stroke patients living in low urbanization areas are expected to have a lower rate of direct visits to stroke units. Similarly, a higher rate of prehospital visits was associated with an increased mortality risk in patients with low urbanization. However, there was no significant difference in the subgroup analysis according to stroke type. Despite efforts to establish acute care hospitals across the country and certified stroke units in rural areas, we confirmed an association between poor prognosis and prehospital visits. Therefore, prehospital visits can be an indicator of health outcomes in the absence of external factors.

This study has two main limitations. First, information about the utilization of an EMS; education levels that can influence stroke awareness; or lifestyle behavior data, such as smoking, exercise, or body mass index, could not be surveyed due to the nature of the claims data. Second, we used health outcomes focused on hospitalization duration, readmission, and mortality and did not include an individual’s functional status or disability. Third, the rate of prehospital visits may have been underestimated when a stroke was not suspected or another diagnosis, such as headache or dizziness, was made because we identified newly diagnosed stroke patients by ICD codes.

## 5. Conclusions

Acute stroke patients with prehospital visits had increased risks of poor health outcomes, such as extended hospital stays, readmission within 180 days, and 30-day mortality, than patients without prehospital visits. The effects of prehospital visits on health outcomes were more pronounced in female patients and those younger than 65 years. Therefore, additional research and policy proposals are needed to clarify the effectiveness of visiting the hospital directly to treat patients with suspected strokes.

## Figures and Tables

**Figure 1 ijerph-20-02488-f001:**
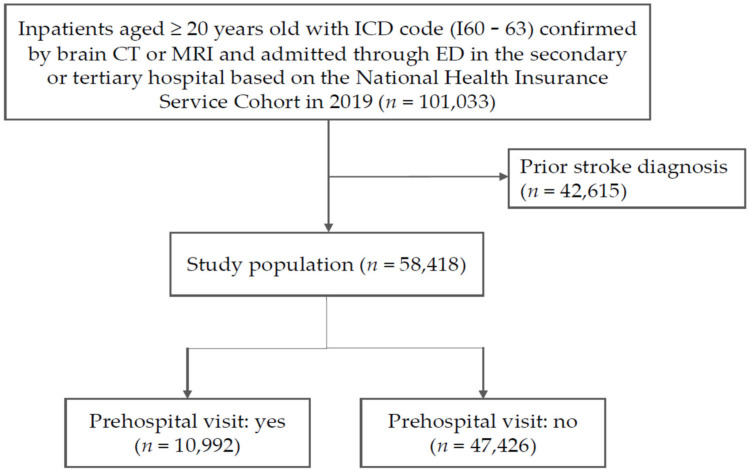
Study population. ICD, International Classification of Diseases; CT, computed tomography; MRI, magnetic resonance imaging; ED, emergency department.

**Figure 2 ijerph-20-02488-f002:**
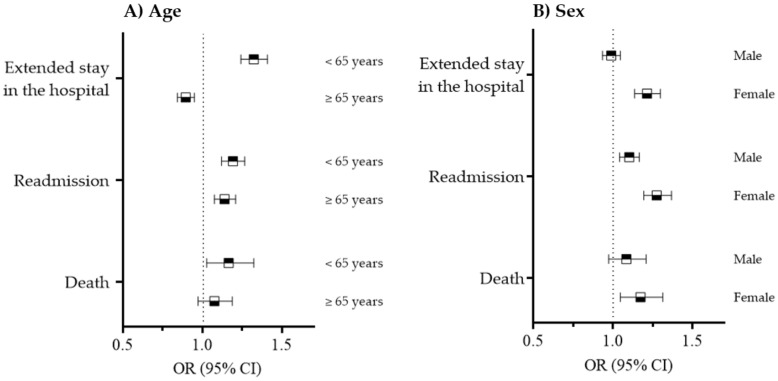
Health outcomes of stroke patients with a prehospital visit compared with patients without a prehospital visit according to age and sex.

**Table 1 ijerph-20-02488-t001:** Demographic and clinical characteristics of acute stroke patients according to prehospital visits.

	Total	Prehospital Visit		*p*-Value
		Yes	No	
*N* (%)	58,418 (100)	10,992 (18.8)	47,426 (81.2)	<0.001
Stroke type				
Ischemic stroke	41,341 (70.8)	6823 (16.5)	34,518 (83.5)	<0.001
Hemorrhagic stroke	17,077 (29.2)	4169 (24.4)	12,908 (76.6)	<0.001
Age, years, mean ± SD	65.9 ± 14.6	65.0 ± 14.4	66.1 ± 14.6	<0.001
<65	26,786 (45.9)	5320 (19.9)	21,466 (80.1)	<0.001
≥65	31,632 (54.1)	5672 (17.9)	25,960 (82.1)	
Sex				0.079
Male	32,938 (56.4)	6280 (19.1)	26,658 (80.9)	
Female	25,480 (43.6)	4712 (18.5)	20,768 (81.5)	
Income status				<0.001
Medical aid	3884 (6.7)	630 (16.2)	3254 (83.8)	
Q1	10,659 (18.3)	2107 (19.8)	8552 (80.2)	
Q2	6972 (11.9)	1296 (18.6)	5676 (81.4)	
Q3	8970 (15.4)	1808 (20.2)	7162 (79.8)	
Q4	1539 (19.8)	2194 (19.0)	9345 (81.0)	
Q5	16,394 (28.1)	2957 (18.0)	13,437 (82.0)	
Urbanization				<0.001
High	37,363 (64.0)	5639 (15.1)	31,724 (84.9)	
Low	21,055 (36.0)	5353 (25.4)	15,702 (74.6)	
Hypertension				<0.001
Yes	28,951 (49.6)	5764 (19.9)	23,187 (80.1)	
No	29,497 (50.4)	5228 (17.7)	24,239 (82.2)	
Ischemic heart disease				<0.001
Yes	5797 (9.9)	951 (16.4)	4846 (83.6)	
No	52,621 (90.1)	10,041 (19.1)	42,580 (80.9)	
Congestive heart failure				<0.001
Yes	4843 (8.3)	808 (16.7)	4035 (83.3)	
No	53,575(91.7)	10,184 (19.0)	43,391 (81.0)	
Diabetes mellitus				<0.001
Yes	4040 (6.9)	660 (16.3)	3380 (83.7)	
No	54,378 (93.1)	10,332 (19.0)	44,046 (81.0)	
Arrhythmia				0.001
Yes	1361 (2.3)	209 (15.4)	1152 (84.6)	
No	57,057 (97.7)	10,783 (18.9)	46,274 (81.1)	

Data are presented as *N* (%) or mean ± standard deviation (SD). *P*-values were obtained using t- and chi-square tests.

**Table 2 ijerph-20-02488-t002:** Health outcomes of acute stroke patients according to prehospital visits.

	Extended Stay in the Hospital	Readmission	Mortality
Total (%)	60.7	52.3	7.48
Prehospital-visit group (%)	62.5	55.3	7.94
No-prehospital-visit group (%)	60.2	51.7	7.38
Odds ratio (95% Confidence intervals)	1.06 * (1.02, 1.10)	1.19 * (1.14, 1.25)	1.23 * (1.13, 1.33)

* *p* < 0.05, in multiple logistic regression analyses.

## Data Availability

The data underlying this article are available in the National Health Insurance Service at https://opendata.hira.or.kr/home.do (accessed on 14 December 2020).

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
