# Peer review of "Association between Prehospital Visits and Poor Health Outcomes in Korean Acute Stroke Patients: A National Health Insurance Claims Data Study"

_ijerph, 2023, doi:10.3390/ijerph20032488_

Round 1

Reviewer 1 Report

Study review: Prehospital visits and health outcomes in newly-diagnosed 2 stroke patients: A national health insurance claims data study 3 in Korea

1. the title does not fully reflect the research carried out. 2.if we are talking about a population study, was the number of n important for a given population calculated? 3.Clarify the aims of the study 4. Table 1 shows the differences between group 1 and group 2. There is no discussion and interpretation of the research. 5. Table 2 does not show statistically significant differences between the groups.

4. There are few references in the discussion to the results of the study and their interpretation in relation to the available studies.

5. The first conclusion resulting from the study should be clarified

Author Response

We are pleased to give us a chance to revise our manuscript and further consideration. We did our best in revising the manuscript following comments raised by the reviewers. We look forward to hearing further. We appreciate the valuable comments.

In addition, the revised manuscript has been proofread by a specialized agency, LangsTech. We attached the certificate. 

Reviewer 1

Study review: Prehospital visits and health outcomes in newly-diagnosed stroke patients: A national health insurance claims data study in Korea.

  1. The title does not fully reflect the research carried out.

Answer) We revised the title to “Association between prehospital visits and poor health outcomes in Korean acute stroke patients: A national health insurance claims data study”.

  1. If we are talking about a population study, was the number of n important for a given population calculated?

Answer) National Health Information Database included the entire Korean population of about 50 million (97.2% of National Health Insurance Service insurers and 2.8% of medical aid). Among them, study subjects (n=58,418) were all patients who were newly diagnosed with stroke in 2019 and whose claims were carried out; ≥ 20 years old who were admitted to the emergency department of secondary or tertiary hospitals and confirmed by brain-computer tomography or magnetic resonance image. We revised lines 86-90 to clarify that the data represents the Korean population (National Health Insurance Service claim data).

  1. Clarify the aims of the study

Answer) We clarified this research aim in lines 61-63.

  1. Table 1 shows the differences between group 1 and group 2. There is no discussion and interpretation of the research.

Answer) We added several explanations related to the differences in characteristics (sex and age in lines 242-248, comorbid disease in 251-254, and urbanization in 263-265) between the prehospital group and the no-prehospital group.

  1. Table 2 does not show statistically significant differences between the groups.

 Answer) The ORs of readmission were increased in the prehospital visit group. However, the ORs of an extended stay in the hospital and mortality after dividing the stroke types were not significantly increased.

The association between prehospital visits and poor health outcomes was confirmed in this group. Furthermore, we aim to compare the different associations according to the stroke types. 

We want to change Table 2 as Reviewer 2’s recommendation to show only these associations in all stroke patients without dividing them to stroke type. Additionally, we added the ‘*’ to show the significant results.

  1. There are few references in the discussion to the results of the study and their interpretation in relation to the available studies.

 Answer) As the first attempt to compare the prehospital visit as a prehospital delay factor, and health outcomes, there was no study using the same index (prehospital visits). We introduced several studies (reference 14, 16-19, and 20) to compare the similar index (ex, onset-to-door time) or health outcomes (ex, reperfusion therapy, anticoagulants, or antiplatelet agents to need an early treatment). In particular, Italian (reference 20) and Japanese studies (reference 14) were introduced in detail because they compared mortality and functional outcome.

  1. The first conclusion resulting from the study should be clarified.

Answer) We are sorry not to understand this question. Does first conclusion mean Abstract’s conclusion? If you let me know again, I will revise it.

Reviewer 2 Report

A brief summary

This study looked at pre-hospital visit and readmission data for 58,418 stroke patients and found that appropriate pre-hospital visits contributed to better health outcomes for patients and proposed pre-hospital visits as a new indicator of pre-hospital delay for stroke patients. The study is an applied study that will help health administrations and providers to optimise diagnostic recommendations and improve the health of stroke patients in the region.

Comment 1

"Introduction" describes the main influences on the efficiency of treatment for stroke patients and suggests that the pre-hospital visit may be one of the influences on the efficiency of treatment for stroke patients. However, the authors do not describe what is included in the pre-hospital visit in Korea, and is it different from other countries? Does the pre-hospital visit also include the investigation of socio-economic and other factors? I would suggest that the authors add the details of the pre-hospital visit in Korea.

Comment 2

In Table 1, the authors categorise pre-hospital visits as 'yes' and 'no'. However, Line 225-227 of the Discussion describes the effect of the number of pre-hospital visits on delayed diagnosis. However, there is no evidence in the results to support this conclusion. I suggest that the authors add the full evidence.

Comment 3

Line 170-171

The authors found that "mortality was higher for pre-hospital visits compared to no pre-hospital visits". However, the reasons for this phenomenon are not explained in the Discussion section. I would suggest that the authors add an explanation.

Comment 4

What are the reasons for the health outcomes of the different stroke types in Table 2? This is not explained in the Discussion. I would suggest that the authors add an explanation.

Comment 5

The labelling of the references is very confusing. I would suggest that the authors make adjustments.

Author Response

We are pleased to give us a chance to revise our manuscript and further consideration. We did our best in revising the manuscript following comments raised by the reviewers. We look forward to hearing further. We appreciate the valuable comments.

In addition, the revised manuscript has been proofread by a specialized agency, LangsTech. We attached the certificate. 

Reviewer 2

A brief summary

This study looked at pre-hospital visit and readmission data for 58,418 stroke patients and found that appropriate pre-hospital visits contributed to better health outcomes for patients and proposed pre-hospital visits as a new indicator of pre-hospital delay for stroke patients. The study is an applied study that will help health administrations and providers to optimise diagnostic recommendations and improve the health of stroke patients in the region.

Comment 1

"Introduction" describes the main influences on the efficiency of treatment for stroke patients and suggests that the pre-hospital visit may be one of the influences on the efficiency of treatment for stroke patients. However, the authors do not describe what is included in the pre-hospital visit in Korea, and is it different from other countries? Does the pre-hospital visit also include the investigation of socio-economic and other factors? I would suggest that the authors add the details of the pre-hospital visit in Korea.

 Answer) We revised the introduction to emphasize the purpose of this study in lines 56-60. In Korea, patients can choose their healthcare providers and institutions because of the low barrier to medical service use and obtain all data of hospital visits from claims. Therefore, this study confirmed the usefulness of prehospital visit as a new indicator reflecting prehospital delay.

Please understand that the prehospital visit indicator is presented for the first time in this study in Korea, and there is no data to indicate the properties of the indicator. 

Comment 2

In Table 1, the authors categorise pre-hospital visits as 'yes' and 'no'. However, Line 234-241 of the Discussion describes the effect of the number of pre-hospital visits on delayed diagnosis. However, there is no evidence in the results to support this conclusion. I suggest that the authors add the full evidence.

 Answer) From the description, it was intended to suggest that pre-hospital visit becomes prehospital delay due to the visit of a number of medical institutions. We would not like to present it as a separate table of figure to compare the results of the effect according to the number of pre-hospital visits. Please understand this. We revised the lines 229-231.

Comment 3

Line 170-171

The authors found that "mortality was higher for pre-hospital visits compared to no pre-hospital visits". However, the reasons for this phenomenon are not explained in the Discussion section. I would suggest that the authors add an explanation.

Answer) This study is to determine the association between prehospital visits and poor health outcomes. Therefore, we have yet to identify the factors to explain why the prehospital visit group has a high mortality. However, as suggested in 210-215, the long hospitalization period and readmission means relatively clinically poor status led to increased mortality.

Comment 4

What are the reasons for the health outcomes of the different stroke types in Table 2? This is not explained in the Discussion. I would suggest that the authors add an explanation.

 Answer) We had thought there would be a difference in the influence of the stroke type on the prehospital visit on the health outcome. However, we want to present the results according to the stroke types as a supplementary table and figure. Reviewer 1 also made a similar recommendation.

Comment 5

The labelling of the references is very confusing. I would suggest that the authors make adjustments.

 Answer) We organized the reference using the Endnote program. Please let me know if you need any additional adjustments.

Round 2

Reviewer 2 Report

Thanks to the author for the explanation of the comments. The article is innovative, but I would suggest that the correlation between the findings and conclusions needs further description.

Author Response

IJERPH-1001367. R2, entitled “Association between prehospital visits and poor health outcomes in Korean acute stroke patients: A national health insurance claims data study”

We are pleased to give us a chance to revise our manuscript again. We try to show the correlation between the findings and conclusions in this revision. We look forward to hearing further.

Reviewer 2

Thanks to the author for the explanation of the comments. The article is innovative, but I would suggest that the correlation between the findings and conclusions needs further description.

Answer) We revised the conclusions in abstract (lines 29-30) and main manuscript (lines 280-285) to show the correlation between the findings and conclusions clearly.

Abstract

pre) Therefore, prehospital visits can be a new index for prehospital delay in patients with acute stroke.

Post) Prehospital visits were associated with unfavorable health outcomes.

Conclusions

Pre) A prehospital visit was associated with poor health outcomes, such as extended hospital stays, readmission within 180 days, and 30-day mortality in Korean patients with acute stroke. The effects of prehospital visits on health outcomes were more pronounced in female patients and those younger than 65 years. Therefore, efforts to visit the hospital directly for treatment are needed for patients with suspected stroke.

Post) Acute stroke patients with prehospital visits had increased risks of poor health outcomes, such as extended hospital stays, readmission within 180 days, and 30-day mortality than patients without prehospital visits. The effects of prehospital visits on health outcomes were more pronounced in female patients and those younger than 65 years. Therefore, additional research and policy proposals are needed to clarify the effectiveness of visiting the hospital directly to treat patients with suspected strokes.